# An ion-electronic hybrid artificial neuron with a widely tunable frequency

Jidong Li [1,2], Wei Zhao[1,3], Chenwei Fu[1,3], Zhenpeng Zhai[1,3], Pengfei Xu[3], Xinyuan Diao[1], Wanlin Guo [1,2] & Jun Yin [1,2] ✉

Biological nervous systems rely on distinct spiking frequencies across a wide range for perceiving, transmitting, processing, and executing information. Replicating this frequency range in an artificial neuron would facilitate the emulation of biosignal diversity but it remains challenging. Here, we develop an ion-electronic hybrid artificial neuron by compactly integrating a nonlinear electrochemical element with a solid-state memristor. This hybrid neuron employing a minimalist architecture exhibits a tunable spiking frequency spanning five orders of magnitude, significantly surpassing the capability of artificial neurons based on electronic devices. Notably, stimuli-dependent ion fluxes enable inherent afferent sensing of liquid flow, temperature, and chemical constituents, eliminating the need for separate, bulky sensors. Connection to biomotor nerves facilitates muscle actuation with frequency-regulated modes. The frequency encoding of a hybrid neuron array allows for the recognition of handwritten patterns. This hybrid neuron design, taking advantage of both ionic and electronic features, offers a promising approach for advanced e-skin and neurointerface technologies.

The spike-based temporal processing in the neural system allows for event-driven, sparse information transfer with high efficiency and reliability[1–3]. While spike amplitude is generally consistent for a given neuron, information is encoded and transmitted through spike frequency and corresponding temporal patterns[4]. These patterns play crucial roles in diverse neural functions, including encoding sensory input intensity and type, motor commands, modulating synaptic plasticity for learning and memory, as well as managing complex cognitive tasks[5–7]. Depending on their specific function within the nervous systems, biological neurons exhibit a broad range of spiking frequencies, from 0.2 to 2 Hz in respiratory neurons to several hundred Hertz in auditory neurons for high-frequency sound response[8].

Mimicking biological spiking neural systems to construct intelligent systems is a long-standing goal, with potential breakthroughs in neurointerfaces, neuromorphic computing, and sensors[9–11]. However, achieving sophisticated control of ionic flow akin to ion channels in biological neurons is far beyond the capabilities of current artificial systems[12–16]. Instead, artificial neurons primarily rely on electronic devices. To mimic the spiking and oscillatory dynamics of neurons, electronic spiking circuits based on conventional transistors consisting of ring oscillators[17,18] or integrate-and-fire devices[19,20] have been widely employed. Although these electronic approaches can reproduce certain aspects of neuronal behavior, the integration of numerous devices leads to complex and bulky circuits, especially when combining separate sensory and spiking circuits for artificial afferent nerves[17,21,22].

Integrating memristors with resistive-capacitive (RC) oscillator circuits to construct artificial neurons can significantly reduce the number of electronic devices, offering the potential for high-density integration[23–28]. The negative differential resistance of memristors drives spiking currents when electrical input stimulation is applied above the threshold value[29]. However, a critical and widespread

[1]State Key Laboratory of Mechanics and Control for Aerospace Structures, Key Laboratory for Intelligent Nano Materials and Devices of the Ministry of Education, Nanjing University of Aeronautics and Astronautics, Nanjing 210016, China. [2]Institute for Frontier Science, Nanjing University of Aeronautics and Astronautics, Nanjing 210016, China. [3]College of Aerospace Engineering, Nanjing University of Aeronautics and Astronautics, Nanjing 210016, China. ✉e-mail: yinjun@nuaa.edu.cn

limitation of these RC-based memristive circuits, and indeed most conventional solid-state artificial neurons, is their typically narrow spiking frequency range, usually spanning only one order of magnitude[19,21,23]. This fundamental constraint severely restricts their ability to accurately emulate the rich diversity of biological signals, which often span multiple orders of magnitude in frequency, and thus limits their utility in complex neuromorphic applications requiring broad dynamic range[8].

Here, we develop an ion-electronic hybrid artificial neuron (HAN) that overcomes these limitations by integrating an ionic electrochemical element in series with an electronic memristor. In contrast to configurations that employ ionic medium as the gates of organic transistors to realize flexible devices[30–33], the design of HAN utilizes the nonlinear behavior of the ion transfer to enable wide and reliable frequency modulation spanning five orders of magnitude. This remarkable tunability far surpasses the capabilities of most existing artificial neurons and is essential for robust rate coding. Furthermore, unlike systems that rely on discrete, external sensors, the stimulus-dependent potential and impedance of ions at the electrochemical interface inherently endow the HAN with direct sensory capabilities. This allows the HAN to function as a compact, integrated afferent nerve for sensing fluid flow, temperature, and chemical information. We connect the HAN to cockroach motor nerves to construct a hybrid bioelectronics reflex arc to actuate muscles. Interestingly, distinguished motion modes can be elicited by altering the spiking frequency, highlighting the advantages of HAN. The frequency encoding of a HAN array also facilitates the recognition of handwritten patterns.

## Results and discussion

Biological neurons maintain a resting potential across the membrane primarily through ion pumps. Upon receiving synaptic input, the neuron's membrane potential changes. When it exceeds a threshold, an action potential, or spike, is generated at the soma[34]. To mimic this spiking behavior of neurons, we developed a HAN with the structure schematically illustrated in Fig. 1a. In contrast to traditional electronic artificial neurons, the HAN comprises an electronic component based on a solid-state memristor and an ionic component based on a liquid electrochemical element.

The $Fe^{2+}/Fe^{3+}$ redox couple within the electrochemical element facilitates the ion-electron exchange between the solid and liquid components with a redox potential of only around ± 0.2 V, as confirmed by cyclic voltammetry (Fig. 1b). The redox reaction mimics the release and reception of neurotransmitters at biological synapses, and thus enables efficient information transfer between the electronic and ionic components. In the electronic component, the $Nb/NbTi_xO_y$ memristor exhibits a threshold switching behavior (Fig. 1c and Supplementary Fig. 1), analogous to the opening and closing behavior of ion channels in biological neurons, thus serving in HAN with a function equivalent to that of biological soma. The current-voltage V(I) and voltage-current I(V) characteristics of the memristor are shown in Fig. 1c. The I(V) characteristic shows that during a voltage sweep, the memristor transitions to a low-resistance state (on) above the threshold voltage ($V_{th}$) and returns to a high-resistance state (off) below the holding voltage ($V_h$). The threshold switching behavior originates from the negative differential resistance characteristic as manifested by the S-shaped V(I) curve. A detailed analysis of its performance is presented in Supplementary Note 1[29].

Combining the ionic and electronic components leads to a HAN with firing capability. Its proposed equivalent circuit is shown in Fig. 1d. The ionic component is represented by the interfacial electric double layer (EDL) capacitance (c), faradic resistance ($Z_f$), and bulk solution resistance. It is worth noting that their impedance is voltage-dependent and varies with time dynamically, as will be discussed later. In the resting state, the memristor maintains a high-resistance state, resulting in low current flow. Upon activation, when the memristor voltage ($V_m$) rises to $V_{th}$, the memristor rapidly switches to a low-resistance state, generating a spike current exceeding 1 mA (Fig. 1e). Subsequently, the interfacial EDL capacitor continues to charge, increasing the voltage drop across the electrochemical element $V_s$ and simultaneously decreasing $V_m$. Once $V_m$ drops to $V_h$, the memristor rapidly transitions back to its high-resistance state. The subsequent gradual discharge of the EDL capacitor leads to an increase in $V_m$, initiating the next cycle. Consecutive I($V_m$) curves of the memristor during HAN firing (Fig. 1f) reveal the EDL charging and memristor switching process within the range between $V_h$ and $V_{th}$ (Supplementary Fig. 2). These switching cycles generate a train of spikes that mimics the biological neurons firing. Importantly, the HANs

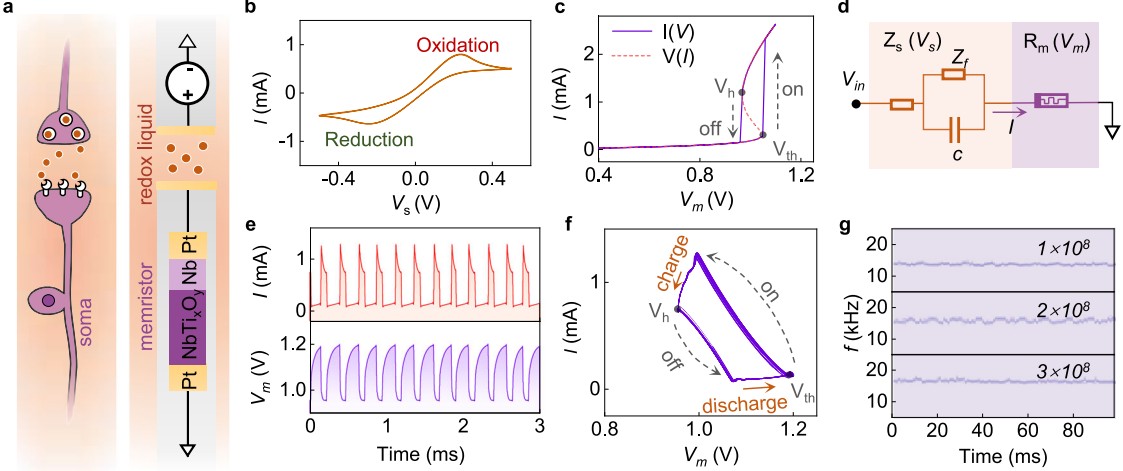

**Fig. 1 | Structure, properties, and stability of hybrid artificial neuron.**
**a** Schematic comparison of a typical biological neuron (left) with our HAN (right). The electronic component of the HAN includes a threshold memristor (purple) and electrodes (gold), while the ionic component (brown) is an electrochemical element with $Fe^{2+}/Fe^{3+}$ redox solution. **b** Cyclic voltammetry curve of the electrochemical element. **c** I(V) (purple solid line) and V(I) (red dashed line) characteristics of the memristor. **d** Equivalent circuit model for the HAN, illustrating the electronic (purple) and ionic (brown) components. **e** Typical firing spikes of the HAN under a constant voltage input, showing changes in voltage (purple) across the memristor and current (red). **f** 33 I($V_m$) curves across the memristor in overlap during the firing of the HAN. **g** Short-time Fourier transform (STFT) map of the HAN's firing activity after $1 \times 10^8$, $2 \times 10^8$, and $3 \times 10^8$ spikes, demonstrating long-term stability with consistent frequency.

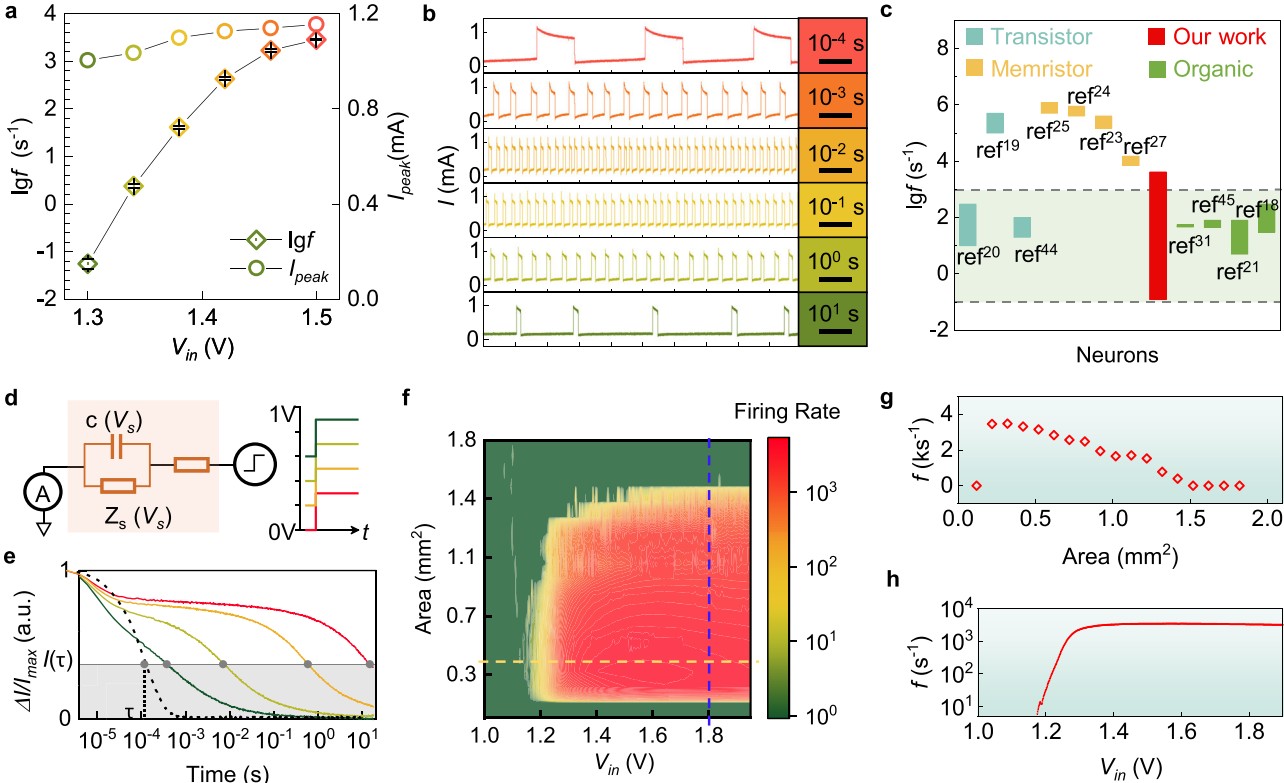

**Fig. 2 | Widely tunable firing frequency of HAN. a** Dependence of the firing frequency and spike peak value on the input voltage. Error bars represent the standard deviation (SD). **b** Spike waveforms corresponding to input voltages in (**a**) with matched colors. **c** Comparison of the frequency range of our HAN with other solid-state ones of different types reported in literature[18–21,23–25,27,31,44,45]. The light green shaded region highlights the working range of human nervous systems. **d** Detailed equivalent circuit diagram of the ionic component (left panel), and the step voltage applied across it for time constant evaluation (right panel). **e** Time evolution of $\Delta I/I_{max}$ under a step voltage of 0.3 V with different baseline voltages. **f** Firing frequency as a function of electrode working area and input voltage. **g** Profile corresponding to the blue dashed line in (**f**), illustrating the typical relationship between the firing frequency and the electrode working area. **h** Profile corresponding to the yellow dashed line in (**f**).

demonstrated stable, consistent spiking behavior over $3 \times 10^8$ cycles (Fig. 1g and Supplementary Fig. 3), indicating excellent durability and suitability for long-term, efficient artificial neural signal transmission and processing.

Surprisingly, the HAN exhibits remarkable frequency tunability over an extensive range during operation. With an input voltage variation ($\Delta V$) of merely 0.2 V, the spike frequency ($f$) spans five orders of magnitude from 0.06 Hz to 2.86 kHz (Fig. 2a), thereby covering the entire spectrum of physiological spike frequencies in the biological nervous system. This corresponds to a voltage-controlled oscillation sensitivity ($\Delta f / f_{min}/\Delta V$) of more than $2.5 \times 10^5$ per volt. While device-to-device variations in the operation voltage were observed, all devices consistently exhibited this wide spiking frequency range (Supplementary Fig. 4). This performance is significantly superior to that of reported solid-state artificial neurons based on transistors, memristors, or organic circuits, whose frequency modulation is mostly confined to a single order of magnitude (Fig. 2c). In these artificial neurons, their firing rate is primarily dominated by the fixed time constant of firing circuit, which is invariant with respect to the input voltage (Supplementary Fig. 5). It results in spiking instability at low input voltages close to the threshold voltage that limits the tunable range of the firing rates (Supplementary Fig. 6 and Supplementary Note 2). Furthermore, a biomimetic feature of the HAN is that the peak amplitude of its spikes remains nearly constant over a large range of firing frequencies (Fig. 2a and b), analogous to the consistent amplitudes of action potentials for a given biological neuron. The energy consumption of the HAN is approximately ~1 mW at a typical firing rate of ~1 kHz. However, by replacing the current Nb/NbTi$_x$O$_y$ memristor

with alternative low-power threshold artificial neurons, such as silicon-based designs or advanced memristors with lower operating currents, we can not only maintain the wide-range frequency tunability but also significantly reduce the energy consumption (Supplementary Fig. 7). This indicates that the principle of wide-range frequency modulation is indeed extensible beyond the specific memristor platform.

To reveal the origin of the HAN's wide frequency tunability, we focus on the dynamic behavior of the ionic component, which distinguishes our HAN from previously reported electronic artificial neurons. During firing events, the memristor operates in a dynamic regime where its voltage rapidly switches between the threshold voltage and the holding voltage (Supplementary Fig. 8a). This intrinsic characteristic of the memristor's switching mechanism leads to a constant voltage difference across the memristor ($\Delta V_m = V_{th} - V_h$) that is independent of $V_{in}$ (Supplementary Fig. 8b). Since the voltage across the electrochemical element is determined by the circuit relationship $V_s = V_{in} - V_m$, the observed lowest solution voltage ($V_S^L$) becomes $V_{in} - V_{th}$ and the highest solution voltage ($V_S^H$) becomes $V_{in} - V_h$. As $V_{in}$ increases, both $V_S^L$ and $V_S^H$ increase linearly, as experimentally confirmed in Supplementary Fig. 8c. Notably, their difference ($\Delta V_s = V_S^H - V_S^L = V_{th} - V_h$) remains equal to the constant $\Delta V_m$. This explains the experimentally observed proportional scaling of $V_S^H$ and $V_S^L$ with $V_{in}$, alongside the invariant $\Delta V_s$, which is an inherent consequence of the memristor's well-defined threshold and holding voltages. Therefore, during the firing of the HAN, the voltage division across the electrochemical element shows a step-like characteristic with a constant voltage step height ($\Delta V_s$) but a DC offset voltage that increases proportionally with $V_{in}$.

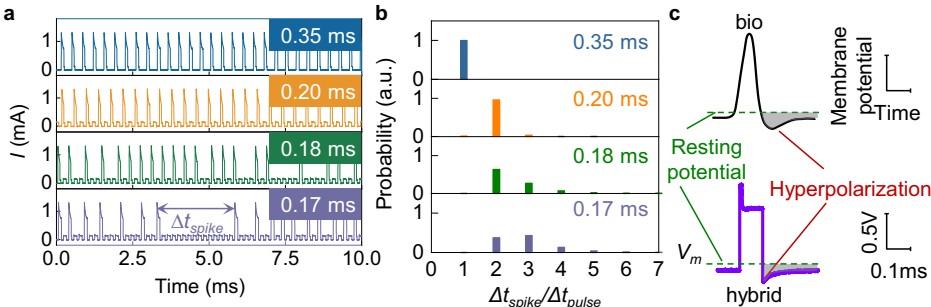

**Fig. 3 | Refractory period in hybrid artificial neurons. a** Firing response of the HAN to different stimulus frequencies. Input voltage pulses were configured with a height of 1.235 V and a 0 V DC offset, maintaining a fixed pulse width of 0.1 ms. The pulse repetition period $\Delta t_{pulse}$ was varied at 0.17, 0.18, 0.20, and 0.35 ms to investigate the frequency-dependent firing dynamics. The resulting spike trains were recorded for each $\Delta t_{pulse}$ condition. **b** Statistical analysis of the firing probability ($\Delta t_{spike}/\Delta t_{pulse}$) for each $\Delta t_{pulse}$ condition, quantifying the average number of input

pulses required to elicit a single spike. A total of 14291 ($\Delta t_{pulse}$ = 0.17 ms), 12351 ($\Delta t_{pulse}$ = 0.18 ms), 11083 ($\Delta t_{pulse}$ = 0.20 ms), and 10132 ($\Delta t_{pulse}$ = 0.35 ms) spikes were recorded for this statistical analysis **c** Voltage response across the memristor (bottom) to the input pulse stimuli in comparison with the action potential of biological neurons (top), showing the hyperpolarization state of the HAN. Gray shaded area indicates the relative refractory region.

The wide frequency tunability of the HAN stems from the highly nonlinear response of the electrochemical element to this dynamically varying voltage. To directly elucidate how this nonlinearity translates into dynamic time constants, step voltages with the same height but different DC offset were applied across the electrochemical element (Fig. 2d). Chronoamperometry results revealed that the time constant of the electrochemical element spanned more than four orders of magnitude, from $10^1$ to $10^{-3}$ s, as the offset voltage was increased from 0 to 0.6 V (Fig. 2e). This effect can be attributed to the accelerated mass transfer of reactive ions in response to the increased potential gradient near the electrode interface. Electrochemical impedance spectroscopy of the electrochemical element further confirms that its impedance is significantly affected by the ion mass transfer within the working frequency region of the HAN (Supplementary Fig. 9 and Supplementary Note 3). Thus, the ion flux in response to the interfacial electrical field that evolves over time leads to highly nonlinear behavior of the HAN, in contrast to electronic circuits, whose behavior can be described simply by the linear R-C model. Ultimately, this unique, widely varied time constant of the electrochemical element with respect to the applied input voltage enables the frequency of the HAN to be widely tuned.

The firing frequency of HANs is not only influenced by the input voltage but also can be significantly tuned by the solid-liquid interfacial impedance. The interfacial impedance plays a crucial role in determining $V_m$. To reveal the underlying relationship, we adjust the contact area between the electrode and the solution, which is the simplest but not the only method to modulate the interfacial impedance. As shown in Fig. 2f, there is a clear demarcation between the spiking state (yellow to red) and resting state (dark green). Spike generation is inhibited when the contact area is less than 0.22 mm² or greater than 1.42 mm². An optimal operating window, where the input voltage and interfacial impedance are appropriately matched (Fig. 2g and h), is necessary to ensure that the voltage variation across the memristor spans the range of $V_h$ to $V_{th}$. This facilitates the non-equilibrium, repetitive threshold switching behavior of the memristor, enabling sustained spiking activity in the HAN. A detailed discussion on load impedance effects is presented in Supplementary Note 1 and Supplementary Fig. 10.

In biological system, it is the refractory period that determines the upper limit of the firing rate of a neuron, preventing the neuron from becoming overexcited[35]. The refractory period is a brief time interval following an action potential, during which the neuron is either incapable of firing or requires a much stronger stimulus to fire[36,37]. To investigate the refractory behavior of the HAN, a series of square-wave pulse sequences with constant pulse width of 0.1 ms but varied periods ($\Delta t_{pulse}$ = 0.35, 0.2, 0.18, and 0.17 ms) were applied as external stimuli, as shown in Fig. 3a. As the stimulus frequency increased, the firing

probability of the neuron decreased. For instance, when the stimulus period was long (0.35 ms or longer), the HAN exhibited stable firing, generating a spike in response to each input pulse. However, with a shorter stimulus period (0.20 ms or less), multiple input pulses were generally required to elicit a single spike, thereby significantly increasing the inter-spike interval ($\Delta t_{spike}$). To quantify this effect, the ratio of $\Delta t_{spike}$ to $\Delta t_{pulse}$ was calculated for more than ten thousand spikes, reflecting the statistical number of pulses required to generate a spike and the firing probability (Fig. 3b).

To reveal the origin of the refractory behavior, the evolution of $V_m$ during the firing process was recorded, as shown in Fig. 3c. Upon activation by an input pulse, the HAN generates a high-current spike, starting to charge the EDL capacitor. Upon termination of the pulse, the memristor enters a hyperpolarization state, i.e. a potential state more negative than the static potential. This hyperpolarized state, induced by the elevated voltage division across the liquid component due to the pre-charged EDL capacitance, persists until the EDL fully discharges. This hinders subsequent spike triggering of the memristor, emulating the refractory behavior in biosystems (Fig. 3c). Notably, even if no spikes are triggered, each input pulse stimulus enhances hyperpolarization (Supplementary Fig. 11), further suppressing subsequent firing. Consequently, as shown in Fig. 3a, more intense stimulus leads to fewer spikes instead.

The nonlinear ion transfer behavior and interfacial impedance are both sensitive to a variety of external stimuli, such as flow motion, temperature, and ion concentrations, which endows the HAN with inherent afferent functionality without the need for additional sensory units (Supplementary Note 4). Based on this, somatosensory functions of mechanosensory[7] (Fig. 4a), thermosensory[38,39] (Fig. 4e), and chemosensory neurons[38,40] (Fig. 4i) have been successfully mimicked using the HAN by converting the analog input signals into the correlated spike frequencies in real time (Supplementary Fig. 12).

To demonstrate mechanosensory capabilities, a microfluidic channel filled with $Fe^{2+}/Fe^{3+}$ redox solution was used as the liquid component of the HAN (Fig. 4b). The flow velocity within the channel changes the rate of mass transfer of redox species to the electrode surface[41], thus could tune the firing rate of HAN. As shown in Fig. 4c, an increase in the flow velocity leads to an increase in the firing rate, which saturates at a flow rate of 33 mm/s. Furthermore, the dynamic response of the neuron to pulsed flow was investigated. Upon the onset of flow, the firing rate increases rapidly and then stabilizes during the constant flow phase. When the flow stops, the firing rate quickly returns to the resting state, demonstrating the capability of real-time detection of flow velocity (Fig. 4d). To the best of our knowledge, this is the first demonstration of an artificial neuron capable of sensing

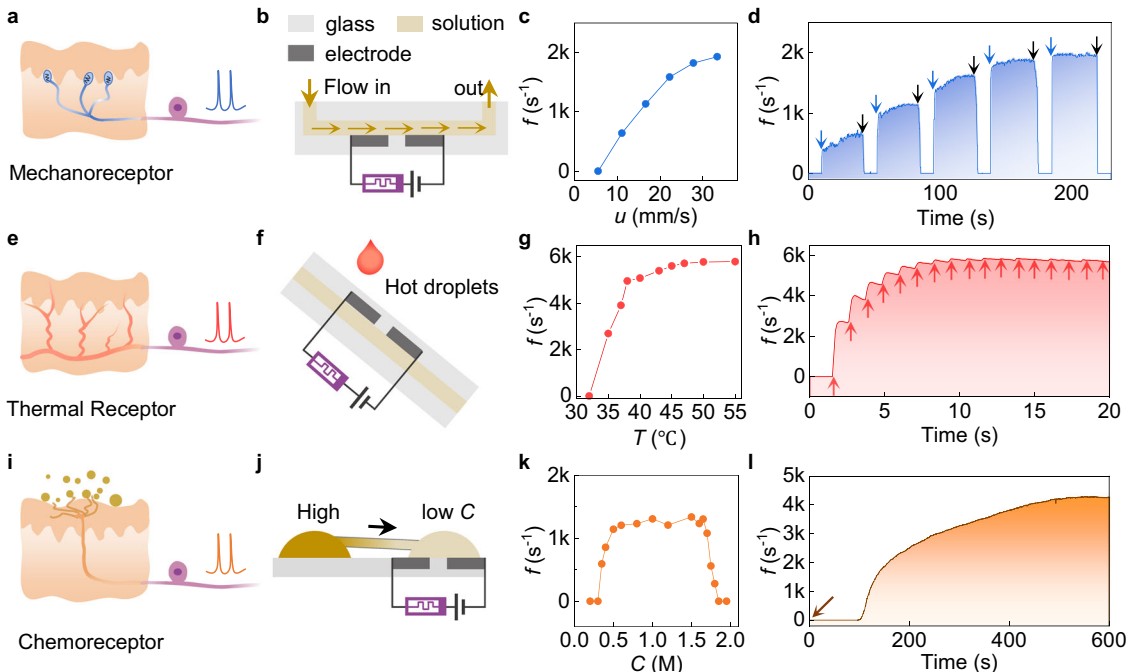

**Fig. 4 | Somatosensory functions of the HAN.** Biomimetic illustration of (**a**) mechanosensory, (**e**) thermosensory, and (**i**) chemosensory neurons. Schematic diagrams of the HAN device structures for sensing (**b**) liquid flow, (**f**) temperature, and (**j**) ion concentration. Dependence of the firing frequency on (**c**) flow rate, (**g**) solution temperature, and (**k**) ion concentration of the solution. Time evolution of the firing frequency during external stimuli of (**d**) pulsed microflow, (**h**) sequential hot droplet impact, and (**l**) concentration-gradient-driven ion diffusion. Blue arrows in (**d**) indicate the onset of flow pulses, and black arrows indicate the termination of flow pulses with flow rates of 11, 17, 22, 28, and 33 mm/s. Red arrows in (**h**) indicate the moment of each hot droplet illustrated in (**f**) impacting on the device surface. Brown arrow in (**l**) indicates the moment the capillary bridge illustrated in (**j**) is established, connecting two droplets of different ion concentrations and initiating diffusion.

liquid flow, opening up exciting possibilities for neuromorphic electronics. Moreover, the mechanism can be extended to ultrasonic vibrations due to a similar mechanism. As shown in Supplementary Fig. 13, the HAN exhibits rapid on/off firing patterns in response to the on/off cycles of an ultrasonic source.

To explore the thermosensory capabilities of hybrid neurons, multiple drops of hot water were sequentially released onto the top surface of the electrochemical element (Fig. 4f), akin to the skin. Once it feels the elevated temperature, the firing rate of HAN increases significantly (Fig. 4g and h). Thermal dissipation after the droplet contacts leads to a reduced frequency, highlighting the real-time response to the surface temperature. The HAN can also be employed as a chemosensor to detect the electrolyte concentration. In the low concentration range from 0.35 to 1.65 M, the spike frequency increases with the concentration and then saturates (Fig. 4k). Further increasing the concentration leads to a reduction in the spiking frequency at an inflection point around 1.7 M. This inhibitory-like response at high stimuli concentrations is analogous to the behavior observed in certain biological neurons, where an excessive stimulus can lead to a decrease in firing rate[42]. Furthermore, the spike waveform is also affected by ion concentration, with the spike width increasing as the ion concentration increases (Supplementary Fig. 14). We demonstrate the capability of HAN to monitor the local electrolyte concentration in real time. Two droplets with different ion concentrations were bridged by a glass capillary, with the HAN monitoring the lower concentration cell (Fig. 4j). The firing rate gradually increases, reflecting the ion diffusion process and the increasing ion concentration of the $Fe^{2+}/Fe^{3+}$ redox couple in the sensing region (Fig. 4l).

Biological systems utilize frequency-encoded neural signals to control various behaviors[21]. Leveraging the widely tunable spiking frequency of the HAN, we explored its potential as an artificial motor neuron to stimulate varied responses of biological systems. As illustrated in Fig. 5a, a HAN was connected to the biomotor nerves of the cockroach's leg, forming a HAN-bio hybrid reflex arc to emulate a biological reflex arc. The flow of information encoded by the spike frequency leads to the actuation of the leg muscles. As shown in Supplementary Fig. 15, the linear sweep of the input voltage from 1.30 to 1.42 V was encoded into spikes with firing rates ranging from ~1 Hz to ~300 Hz. It is surprising to observe that spiking of different frequencies could trigger the leg's motion of distinguished modes. Firing at a rate of approximately 180 Hz induced a back-and-forth swinging motion of only the cockroach's claws (Fig. 5b). Reducing the frequency to 120 Hz successfully stimulated tarsal swing (Fig. 5c), while further decreasing it to around 10 Hz induced a pronounced contraction and rotation of the tibia (Fig. 5d). These results demonstrate the capability of HAN to control the diverse motion modes of biosystem through largely turned frequency during operation.

Integration of multiple HANs to form an array enables sensory perception and pattern recognition. We constructed a thermosensory array capable of sensing finger touch and recognizing the corresponding handwritten digits. As illustrated in Fig. 5e and Supplementary Fig. 16, the $Fe^{2+}/Fe^{3+}$ redox solution was confined between two ITO glass plates with a ~100 μm gap in between. Adopting this configuration, the HAN array can be easily fabricated by patterning the top ITO glass with designed numbers, thus facilitating compact integration of large scale. The output spikes from all HANs were summed by the bottom ITO plate and recorded by an ammeter, mimicking the function of a post-synapse in connection with multiple biological synapses adding action potentials from multiple presynaptic neurons. We patterned the top ITO electrode into a 3 ×2 array to serve as the sensory interface for six individual HANs labeled as $N_{ij}$. The area of the bottom ITO electrode is more than one order of magnitude larger than the sum of the working electrode areas of all individual HANs, ensuring negligible interfere of the neighboring HANs (Supplementary Fig. 17).

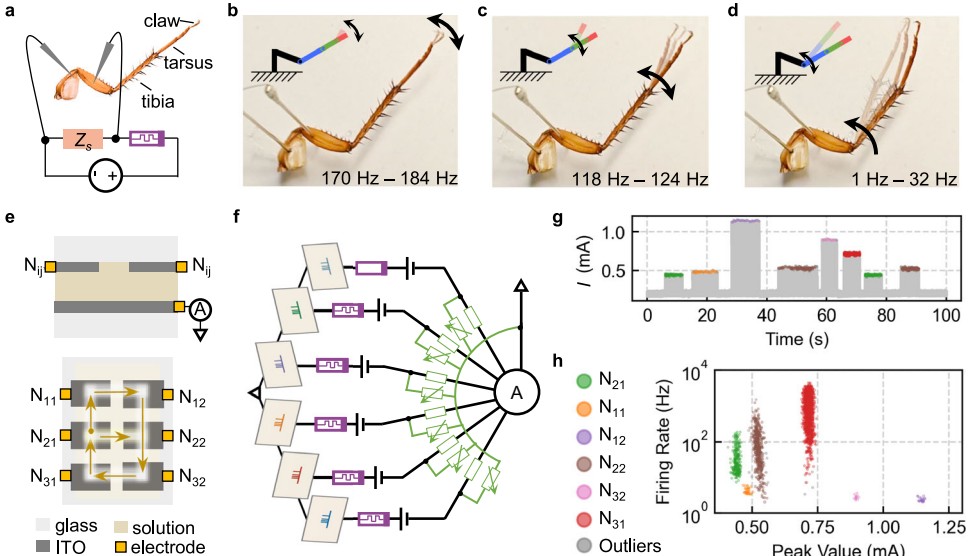

**Fig. 5 | HAN for neurointerfaces and handwriting recognition. a** Schematic illustration of a detached cockroach leg with its efferent nerve connected to the HAN for bioelectronic interfacing. **b**–**d** Response of the cockroach legs to the spiking stimulation of different frequencies. **e** Schematic diagram of the HAN sensing array for handwriting recognition. The upper and lower panel presents a cross-sectional view and a top view of the array structure, respectively. The arrows in the lower panel indicate a representative trajectory for the handwritten digit 8, starting from the $N_{21}$ electrode. **f** Circuit of an array consisting of 6 hybrid artificial neurons for handwritten digit recognition. **g** Output of the sensing array to the handwriting of 8. **h** Distinguished peak currents and firing rates of these HANs.

In biological systems, synaptic weight is another key factor that determines how information is encoded, transmitted, and processed in the brain. To emulate the synaptic weight, a pair of shunt resistors were connected in series with each neuron to adjust its spike amplitude and thus the weight (Fig. 5f). Benefiting from the nearly constant spike amplitude independent on stimuli intensity, the postsynaptic spike amplitude can be assigned to specific neuron as a recognition feature. The spike frequency could be tuned by adjusting the input voltage or top electrode area of each neuron. In this way, the firing signal can be dual-encoded by assigning to each neuron spiking characteristics in terms of distinct frequency and amplitude.

Figure 5g shows the summed spike train corresponding to the handwritten 8 input trace illustrated in Fig. 5e. Analysis of the temporal profiles of the spiking signal demonstrates the possibility of a bioinspired way to recognize the handwritten pattern. Using a DBSCAN clustering algorithm (Supplementary Note 5), each detected spike within the recorded train was automatically sorted into six valid clusters, based on its unique frequency and amplitude characteristics (Fig. 5h). This clustering enables the identification of which HAN generated a particular spike. For instance, the cluster characterized by a frequency spanning from ~8 to 162 Hz and a minimum spike amplitude of ~0.44 mA is assigned the color green and corresponds to the starting neuron ($N_{21}$) on the writing path of the handwritten 8. In contrast, the cluster characterized by the lowest frequency around 2.4 Hz but the highest spike amplitude around 1.1 mA is assigned the color purple and corresponds to neuron $N_{12}$. The final recognition decision for the handwritten digit is determined by the temporal sequence of these identified clusters. By assigning cluster colors back to each spike's peak in the temporal waveform (Fig. 5g), the spatial trajectory of the handwritten 8 can be visualized, revealing the activated sequence of neurons: $N_{21}$, $N_{11}$, $N_{12}$, $N_{22}$, $N_{32}$, $N_{31}$, $N_{21}$, and $N_{22}$. Recognition of other handwritten digits is presented in Supplementary Fig. 18, further demonstrating the feasibility of using the HAN array for pattern recognition.

Notably, neurons $N_{21}$, $N_{22}$, and $N_{31}$ exhibit a broader spike frequency range compared to the other neurons, as shown in Fig. 5h. This phenomenon is directly attributed to their much higher intrinsic firing rates combined with the dynamic thermal transfer effect during finger

contact and release. For instance, after the finger leaves a specific region, the temperature gradually decreases, causing the firing rate to consistently drop from a high level to a lower level, as shown in Supplementary Fig. 19. This extended period of decreasing firing rate, rather than an abrupt cessation, manifests as a broader spectrum of observed frequencies for these sensitive neurons. In contrast, neurons such as $N_{11}$, $N_{32}$, and $N_{12}$ possess much lower intrinsic firing rates, resulting in narrow distributions, highlighting the high degree of designability of the HAN.

HAN is fundamentally based on the controllable impedance of the $Fe^{2+}/Fe^{3+}$ redox solution, which offers several distinct advantages for advanced neuromorphic systems. It enables an exceptionally broad frequency tunability spanning five orders of magnitude, inherently provides direct sensing capabilities for various physical and chemical stimuli without external transducers, and mimics the dynamics of biological ion channels. While the current proof-of-concept HANs employ arrays of millimeter-scale electrochemical cells, scalability is crucial for future high-density neuromorphic applications. Miniaturization of the electrochemical component can be achieved by reducing the operating current of HANs. This can be realized by replacing the $Nb/NbTi_xO_y$ memristor with lower-power threshold elements (Supplementary Table 1). For instance, transistor-based HANs, as demonstrated in Supplementary Fig. 7, enable electrode areas to be reduced to micrometers, and are also compatible with standard CMOS processes for further miniaturization of the solid-state component. Furthermore, increasing the effective specific surface area of the electrodes, for instance, through electroplating of platinum black, allows a substantial reduction in physical electrode size while maintaining the required working current (Supplementary Fig. 20). Beyond individual device miniaturization, fabricating larger HAN arrays necessitates advanced integration strategies. Three-dimensional integration techniques enable vertical stacking of multiple HAN devices, thereby significantly increasing device density without expanding the overall footprint. Additionally, optimizing internal structure through advanced packaging technologies such as Through Glass Via (TGV) or Flexible Printed Circuits (FPC) ensures efficient isolation of ionic and electronic components, enabling compact and even flexible integration (Supplementary Fig. 21).

In summary, by taking advantage of the nonlinear ion transfer process of electrochemical element and robust threshold switching of solid-state memristor, we have successfully demonstrated an ion-electronic hybrid artificial neuron capable of mimicking key neuronal behaviors, including action potential generation, refractory period, and spike frequency regulation. Especially, the firing frequency of the neuron exhibits a wide-range tunability over five orders of magnitude, unprecedented for traditional solid-state artificial neurons, fully covering the working range of biological nervous systems. The sensitivity of ion flux to environmental stimuli such as flow rate, temperature, and ion concentration endows it with the inherent capability to emulating mechanoreceptors, thermoreceptors, and chemoreceptors without the need to integrate additional bulky sensors. The hybrid artificial neurons have been successfully applied in response-mode-designed electrophysiological stimulation and handwritten recognition, demonstrating its promising potential in neural interface and bioinspired sensor technologies.

## Methods

### Memristor fabrication

Threshold memristors were fabricated by sequential sputtering deposition of four layers with designed patterns on silicon substrates with a 285 nm $SiO_x$ insulating layer. The patterns were defined by direct projection UV photolithography (Tuotuo Technology TTT-07-UV Litho-ACA Pro) on AZ5214 photoresist (Suzhou Juxin Mems Technology Co., Ltd). A 25 nm thick bottom Pt electrode was firstly fabricated. Subsequently, a top electrodes consisted of $NbTi_xO_y$ (60 nm), Nb (40 nm), and Pt (25 nm) layers were deposited to form a crossbar structure with the bottom electrode (Supplementary Fig. 22)[43]. To introduce threshold switching behavior, an electroforming process was performed by applying a voltage exceeding 7 V to the prepared memristor.

### Electrochemical element fabrication

The electrochemical element consists of ITO electrodes and $Fe^{2+}/Fe^{3+}$ redox couple solution. The redox solution was prepared by dissolving 1 M $FeCl_2 \cdot 4H_2O$ (Sinopharm Chemical Reagent Co., Ltd) and 1 M $FeCl_3 \cdot 6H_2O$ (Aladdin) in deionized water, unless otherwise stated. The configuration of electrochemical elements was customized for different applications. For mechanosensory and thermosensory, the redox solution was injected into a microchannel. The microchannel was constructed by placing two ITO glass substrates face-to-face separated by a 100 μm thick PDMS film with a rectangular groove in the center of the film. Prior to assembly, both the PDMS and ITO glass surfaces were treated with oxygen plasma to enhance interfacial adhesion. For chemosensory, a 100 μm thick PDMS layer with two 3 mm diameter holes was used to cover the ITO glass, forming an exposed window for contacting with the redox solution. Fork-finger electrodes (for mechanosensory) or pad electrodes (for thermosensory and chemosensory) were patterned on ITO-coated glass substrates using laser etching (Foster Fiber Laser Marking Machine). Refer to Supplementary Note 4 for details.

### Electrical and electrochemical characterization

Input voltage stimuli were generated using an arbitrary function generator (Tektronix AFG31052). Real-time voltage waveforms were detected using a high-resolution oscilloscope (Rigol DHO4804). Current signals were recorded by the oscilloscope coupled with a low-noise preamplifier (Stanford Research SR570). The simultaneous voltage and current measurements were performed using a two-channel oscilloscope with a combined current preamplifier, as depicted in Supplementary Fig. 23. Current is measured via the current preamplifier, whose output, proportional to current, is connected to one channel of the oscilloscope. Another oscilloscope channel with a high-impendence (10 MΩ) probe is directly employed to measure the memristor voltage. The I-V characteristics of the threshold memristors were measured using a Keithley 2450 source meter. The electrochemical elements were analyzed by cyclic voltammetry and electrochemical impedance spectroscopy using an electrochemical workstation (Princeton P3000A-DX). To map the firing frequency as a function of both input voltage and electrode area, a stepper motor was used to control the immersion depth of the rectangular ITO electrodes into the redox solution. The firing activity was measured at each depth of immersion by sweeping the input voltage.

## Data availability

The source data supporting the findings of this study are available in Figshare at https://doi.org/10.6084/m9.figshare.29528465.

## Code availability

The code supporting the DBSCAN clustering are available in Figshare at https://doi.org/10.6084/m9.figshare.29528465.

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

## Acknowledgements

This work was supported by National Key Research and Development Program of China (No. 2024YFA1409600, J. Y.), National Natural Science Foundation of China (T2293691, W. G.; No. 12372329, J. L.; T2293694, J. L.; 12172176, J. Y.; 12311530052, J. Y.), Natural Science Foundation of Jiangsu Province (No. BK20243065, W. G.), the Research Fund of State Key Laboratory of Mechanics and Control for Aerospace Structures (No. MCAS-I-0125Y01, J. Y.; MCAS-I-0525K01, W. G.), the Fundamental Research Funds for the Central Universities (No. NC2023001, W. G.; NJ2023002, W. G.; NJ2024001, W. G.) and the Fund of Prospective Layout of Scientific Research for NUAA (Nanjing University of Aeronautics and Astronautics).

## Author contributions

J. L., J. Y. and W. G. conceived the project. With assistances from J. L., W. Z., C. F., Z. Z. and X. D. performed the experiments. P. X. and J. L. performed data clustering analysis. J. L., W. Z. and J. Y. wrote the paper. All the authors discussed the results and reviewed the manuscript.

## Competing interests

The authors declare no competing interests.
