## [Transparent Peer Review file · Nature Communications]

An ion-electronic hybrid artificial neuron with a widely tunable frequency

Corresponding Author: Professor Jun Yin

Version 0:

Reviewer comments:

Reviewer #1

(Remarks to the Author)

In the paper “An ion-electronic hybrid artificial neuron with a widely tunable frequency,” the authors report that the combined use of ionic and electronic properties can effectively emulate the behavior of biological neurons. In particular, the ability to achieve rate coding through spike frequency modulation, along with the implementation of a biological-like refractory period that mimics the brain’s mechanism for preventing over-firing, presents a highly intriguing possibility for many researchers working in this field. However, the scalability and generalizability of the proposed hybrid neuron, as well as its suitability for computational applications, require further demonstration. Therefore, I would prefer to assess the suitability of this manuscript for publication in Nature Communications after the following revisions are addressed. The following are detailed suggestions:

1. In Figure 2c, the HAN successfully mimics the broad frequency range of biological neurons. While achieving biological realism is valuable, throughput is also a critical factor for applications such as accelerated neural network inference and data-heavy tasks. In this regard, is it possible to further extend the firing rate range to enable faster operation, or is it fundamentally limited by the ionic component?
2. Given that the electrochemical element plays a dominant role in spike frequency modulation, can this approach be extended to other mechanisms, such as RRAM-based or similar artificial neurons?
3. What is the device-to-device variability of the HAN? A more detailed analysis is recommended.
4. The electrode working area (~mm²) is considerably large. The authors can discuss more about how to design and fabricate larger device arrays for future high-density applications, as well as how to achieve integration of the internal structure within the HAN.
5. On page 7, refractory-induced hyperpolarization—relevant to the prevention of over-firing in the brain—is an important inhibitory characteristic of biological neurons. A reference should be provided to help readers better understand this biological mechanism. In addition, a citation is also needed to support the statement on page 9, line 228: “It is similar to the case of biological neurons, whose firing rate would decrease for noxiously high stimuli exhibiting protective inhibition.”
6. In Supplementary Note 2, the explanations in line 66 (firing rate fluctuations with respect to V_{in}) and line 72 (exponential decay due to the time constant) refer to Fig. S3b and Fig. S3c, respectively. However, the figures do not appear to match the corresponding descriptions. Clarification is needed to ensure consistency between the text and the referenced figures.
7. Fig. S9 is not mentioned or discussed in either the main manuscript or the Supplementary Information. An explicit explanation should be provided to ensure a coherent logical flow for the reader.
8. In Fig. 4d, h, and i, the transient firing frequency responses are presented to demonstrate the somatosensory functions of the HAN. Providing clearer labeling and more descriptive captions would enhance reader comprehension. In addition, it would be helpful to clarify the type of view (e.g., top view, cross-sectional view) presented in Fig. 5e.
9. In Fig. S12, although an identical ultrasonic source is applied, the spike density appears to increase with repeated stimulation. Does the HAN maintain a constant firing rate in response to consistent input, or does this reflect a form of short-term learning behavior?
10. The caption of Fig. S15 refers to subfigures a–f; however, only a–d are shown in the figure. This discrepancy should be corrected to avoid confusion.

(Remarks on code availability)

Reviewer #2

(Remarks to the Author)

This study presents a hybrid artificial neuron (HAN) that integrates a solid-state memristor with an ionic electrochemical element, achieving wide frequency tunability and inherent sensory capabilities. The work addresses critical limitations of conventional solid-state artificial neurons, such as narrow frequency ranges and reliance on external sensors, by leveraging the nonlinear dynamics of ionic-electronic interfaces. The concept is innovative and holds promise for neuromorphic computing applications. However, the following points should be addressed before the manuscript can be considered for acceptance:

1. The manuscript should specify the parameters of the input voltage used throughout the spiking experiments, including pulse height, interval, width, and total number of pulses.
2. Please provide a detailed description of the experimental setup and methodology used to acquire the data presented in Figure 1e. Additionally, clarify the rationale for representing the time scale in milliseconds ("ms").
3. Figure S6 shows the voltage response across the solution during firing events. As the input voltage increases, both VSH and VSL increase proportionally, while the voltage difference remains nearly unchanged. The authors are asked to provide an explanation for this behavior.
4. In Figure S10, the authors present the refractory period of the HAN. However, it is unclear why a membrane potential (V_m) is observed without a corresponding current spike at $T = 0.25$ ms and 0.15 ms. Furthermore, clarification is needed regarding the initial overshoot and subsequent stabilization of V_m during the current spiking event.
5. Regarding the array configuration shown in Figure 5e, please specify the exact dimensions of the setup. The manuscript should also provide evidence supporting the claim that neighboring cells do not interfere with each other, especially considering that all cells share the same electrolyte and bottom electrode.
6. For handwriting recognition, the authors claim spike amplitude can be assigned to specific neurons as a recognition feature. However, it is unclear how the final recognition decision is determined from the network's output. In Figure 5h, it would be helpful to explain clearly why neurons N21, N22, and N31 exhibit a broader spike frequency range compared to the other neurons.
7. What is the energy consumption of the HAN system? The authors should report and compare the energy consumption of HAN with other neuron memristors.
8. The broadly tunable frequency of HAN is fundamentally based on the controllable impedance of the Fe^{2+}/Fe^{3+} redox solution. The authors should elaborate on the advantages of this redox approach and discuss its impact on the scalability and integration of HAN circuits, particularly given the complexities associated with incorporating liquid-phase redox systems in compact architectures.
9. The authors are encouraged to include TEM or SEM images to illustrate the morphology of the threshold memristor.

(Remarks on code availability)

Reviewer #3

(Remarks to the Author)

(Remarks on code availability)

Reviewer #4

(Remarks to the Author)

In this manuscript, authors developed an ion-electronic hybrid artificial neuron by compactly integrating a nonlinear electrochemical element with a solid-state memristor, which exhibits a tunable spiking frequency spanning five orders of magnitude, significantly surpassing the capability of artificial neurons based on electronic devices. This hybrid neuron design, taking advantage of both ionic and electronic features, offers a promising approach for advanced e-skin and neurointerface technologies. This is an innovative research work and the paper is well written. Therefore, I recommend that this paper can be accepted for publication after appropriate revisions.

- (1) The author should compare previous relevant reports to highlight the uniqueness of this study in the Introduction section.
- (2) Lack of discussion on the working mechanism of the device, the author should supplement relevant discussions in the main text.
- (3) What is the thickness of the functional layer? The author should have provided a cross-sectional SEM image.
- (4) Can the author present the I-V curve of a symmetrical voltage window in the main text or supporting information.
- (5) Grammar errors require careful correction by the author.

(Remarks on code availability)

Version 1:

Reviewer comments:

Reviewer #1

(Remarks to the Author)

The authors have mostly addressed the reviewer's concerns.

(Remarks on code availability)

Reviewer #2

(Remarks to the Author)

The authors have adequately addressed all my previous comments. I have no further concerns.

(Remarks on code availability)

No specific question.

Reviewer #3

(Remarks to the Author)

(Remarks on code availability)

Reviewer #4

(Remarks to the Author)

This manuscript can be accepted for publication.

(Remarks on code availability)

Reply to Referee #1:

General Comments: *In the paper “An ion-electronic hybrid artificial neuron with a widely tunable frequency,” the authors report that the combined use of ionic and electronic properties can effectively emulate the behavior of biological neurons. In particular, the ability to achieve rate coding through spike frequency modulation, along with the implementation of a biological-like refractory period that mimics the brain’s mechanism for preventing over-firing, presents a highly intriguing possibility for many researchers working in this field. However, the scalability and generalizability of the proposed hybrid neuron, as well as its suitability for computational applications, require further demonstration. Therefore, I would prefer to assess the suitability of this manuscript for publication in Nature Communications after the following revisions are addressed. The following are detailed suggestions:*

General Reply: We sincerely appreciate the referee for recognizing the significance of our work and providing insightful feedback. To comprehensively address the concerns regarding the scalability and generalizability of our proposed hybrid artificial neuron (HAN), we have conducted additional experiments as discussed in replies to Comment 2 and 4, and made revisions to the manuscript.

Regarding the suitability for computational applications, the HAN's wide-range frequency tunability, spanning five orders of magnitude, makes it an exceptional candidate for highly efficient, logarithmic rate-encoded units in spiking neural networks (SNNs). Unlike the inherently stochastic nature of Poisson encoding, where spike times are random at an average rate, our logarithmic rate coding provides a deterministic and reproducible relationship between the input voltage and the resulting firing rate. It also allows for easier decoding compared to Poisson encoding, which often requires extensive averaging of spike trains, a particularly challenging aspect for low-rate inputs. This behavior may simplify SNN design and training.

Fig. R1 | Firing rate dependence on the input for the HAN. The experimental firing rate data of the HAN is accurately fitted using a modified Exponential Linear Unit (ELU) function. This piecewise function describes the relationship between the input (x , representing the input voltage) and the firing rate (y):

$$y = \begin{cases} y_0 + \alpha \cdot (1 - \exp(-k \cdot (x - x_0))) & \text{if } x \geq x_0 \\ y_0 + k(x - x_0) & \text{if } x < x_0 \end{cases}$$

where x_0 , y_0 , α , and k are fitting parameters derived from the data.

Furthermore, the inherent non-linear relationship between input and firing rate of the HAN is a powerful feature, potentially serving as the neuron's activation function. This relationship is accurately fitted by a modified Exponential Linear Unit (ELU) function (**Fig. R1**), a form of activation function commonly employed in SNNs. Such non-linearity is critical for enabling the network to

learn and approximate complex, non-linear functions of its inputs, allowing single neurons or small ensembles to perform sophisticated transformations and extract intricate features. Consequently, this can lead to more compact and computationally efficient SNN architectures.

Moreover, the HAN's intrinsic somatosensory functions make it suitable for direct sensor-to-compute applications at the edge. Edge computing involves processing data locally on devices or at the periphery of the network, minimizing reliance on centralized cloud servers. HAN's ability to directly convert physical and chemical stimuli into neural spikes can potentially eliminate the need for bulky and power-intensive analog-to-digital converters, thereby simplifying the data process from sensing to computation.

Comment 1: *In Figure 2c, the HAN successfully mimics the broad frequency range of biological neurons. While achieving biological realism is valuable, throughput is also a critical factor for applications such as accelerated neural network inference and data-heavy tasks. In this regard, is it possible to further extend the firing rate range to enable faster operation, or is it fundamentally limited by the ionic component?*

Reply 1: We acknowledge that the upper limit of our HAN's firing rate is indeed primarily constrained by the ionic component, specifically the charging and discharging time constant (τ) of the electric double layer (EDL). This time constant is determined by the product of the interfacial faradic resistance (R_t) and the EDL capacitance (C_{dl}), given by $\tau = R_t \times C_{dl} = \rho \times c$, where ρ and c are the faradic resistance and the EDL capacitance per unit area. Thus, to enable faster operation, ρ and c need to be reduced.

For instance, by replacing the ITO electrodes with platinum (Pt) electrodes, we successfully increased the spiking rate from approximately 5 kHz to ~20 kHz (**Fig. R2**). This improvement is attributed to the faster electrode reaction rate of Pt, which significantly decreases ρ . However, further extending the firing rate for extremely high-throughput applications remains a challenge. Potential approaches for future work include increasing the faradic current density through the employment of catalytic layers, optimizing electrolyte composition for faster ion reactivity, or exploring alternative electrochemical systems with intrinsically lower time constants.

Fig. R2 | Spiking responses of HAN utilizing Pt as working electrode. a, Dependence of the firing frequency on the input voltage. **b,** Spike waveforms corresponding to input voltages in (a) with matched colors.

Comment 2: Given that the electrochemical element plays a dominant role in spike frequency modulation, can this approach be extended to other mechanisms, such as RRAM-based or similar artificial neurons?

Reply 2: We appreciate the reviewer for this insightful comment. Our findings indicate that the principle of wide-range frequency modulation, driven by the voltage-dependent time constant of the ionic component, is indeed extensible beyond the specific memristor platform. To explore this versatility, we constructed a transistor-based HAN (**Fig. R3a-c**). This alternative configuration also demonstrated a broad frequency range (**Fig. R3d, e**), confirming generalizability of our proposed mechanism for the hybrid artificial neuron. Moreover, owing to the ultralow ON current characteristics of the transistor-based HAN, this configuration exhibits a significantly reduced working current of $\sim 3 \mu\text{A}$, consequently, permitting a much smaller working electrode area (**Fig. R3f-h**). This reduction in electrochemical cells, coupled with the inherent scalability of transistor fabrication via CMOS processes, holds significant promise for facilitating high-density integration of HAN arrays in future neuromorphic systems, as we will discuss in the following reply to Comment 4.

Fig. R3 | Transistor-based HAN exhibiting widely tunable firing frequency, low power consumption, and microelectrode. **a**, Circuit schematic of transistor-based artificial neuron unit consisting of PNP and NPN transistors and two resistors. **b**, I_m - V_m characteristics of the transistor-based neuron exhibiting threshold switching behavior, similar to Nb/NbTi_xO_y memristor but with a much lower ON current of $\sim 3 \mu\text{A}$. **c**, Schematic circuit of a HAN, composed of transistor-based neuron (purple element) in serials with a liquid cell (brown element). **d**, Spike waveforms corresponding to different input voltages V_{in} corresponding to that of matched colors in (e). **e**, Log dependence of the firing frequency f on V_{in} , exhibiting a widely tunable firing frequency up to 5 orders of magnitude. **f**, Optical image of four transistor-based HANs integrated on one PCB. **g**, Magnified image of the red dashed rectangular region in (f), showing the electrochemical liquid cell part of HANs with four-channel Pt electrodes and a PDMS microchannel. **h**, Magnified image of the red dashed rectangular region in (g), exhibiting the Pt microelectrodes (white) and the PDMS microchannel of redox solution (Yellow).

Comment 3: What is the device-to-device variability of the HAN? A more detailed analysis is recommended.

Reply 3: To evaluate the device-to-device variability, we fabricated and characterized 10 HAN devices, measuring their spiking rate as a function of the input voltage (V_{in}). As shown in **Fig. R4a**, all devices consistently exhibited a wide spiking frequency range, from over 1 kHz down to approximately 0.01 Hz. However, we observed variations in the specific operating voltage required to achieve a particular spiking rate (e.g., 100 Hz), which ranged from 1.15 V to 1.65 V, with an average of 1.44 V. This variability primarily stems from the performance variations in the memristor component of the HAN, as evidenced by the dispersion in the I-V curves of the memristors (**Fig. R4b, c**). In contrast, the electrochemical liquid cell components demonstrated good consistency, as indicated by the highly reproducible cyclic voltammetry results (**Fig. R4d**). The variability of the memristor properties need further optimization to improve the overall uniformity of HAN device performance for large-scale integration, but it remains challenging (*Science*, 2023, 381: 1205). Alternatively, our transistor-based HANs (**Fig. R3**) show good consistency due to their much better uniform threshold behavior (**Fig. 4e, f**).

Fig. R4 | Device-to-device variability of the NbO_x-based HAN (a-d) and transistor-based HAN (e,f). **a**, Log dependence of the firing frequency f on V_{in} of 10 HANs. **b**, I-V characteristics of the 10 HANs' memristor part, exhibiting obvious device-to-device variation. **c**, Statistical variation of the threshold voltage (V_{th}) and holding voltage (V_h) of the 10 memristors. **d**, Cyclic Voltammetry of the 10 HANs' electrochemical cell part, exhibiting small variation. **e**, I-V characteristics of memristor part of 4 transistor-based HANs', exhibiting minor device-to-device variation. **f**, Log dependence of the firing frequency f on V_{in} of 4 transistor-based HANs, indicating minor device-to-device variation.

Comment 4: The electrode working area ($\sim mm^2$) is considerably large. The authors can discuss more about how to design and fabricate larger device arrays for future high-density applications,

as well as how to achieve integration of the internal structure within the HAN.

Reply 4: We understand the importance of achieving larger device arrays for high-density applications and acknowledge the large electrode working area adapted in this work. We propose several strategies and incorporate the corresponding discussion in the main text.

1. Reducing the working current and electrode area: The working current of the HAN is primarily determined by the solid-state memristor. As demonstrated in our response to Comment 2 and **Fig. R3**, by replacing the NbTi_xO_y memristor with other threshold artificial neurons exhibiting lower working current such as silicon-transistor-based neurons, the working electrode size can be significantly reduced to micrometers or even sub-micrometer scales in principle. Such transistor-based neurons are highly scalable through standard CMOS processes. This reduction in working current directly leads to the miniaturization of the electrochemical cell.

2. Increasing the specific surface area of the electrode: We have explored methods to increase the effective specific surface area of the electrodes. For instance, electroplating platinum black (Pt-black) onto Pt electrodes significantly increased the effective surface area, leading to a ~ 2.5 -fold decrease in interfacial impedance (**Fig. R5**). This decrease in impedance directly implies that the electrode footprint can be correspondingly reduced while maintaining the same working current, thereby enabling further miniaturization.

Fig. R5 | Characterization of platinum black (Pt-black) electrodes. **a**, SEM image of Pt wire electroplated Pt-black. **b**, Magnified SEM image corresponding to the red box region in **(a)**. **c**, Bode plot of electrochemical impedance. The plot illustrates a substantial decrease in the electrochemical impedance for the Pt-black electrode compared to a bare Pt electrode of the same nominal footprint, particularly within the low-frequency region where the impedance is dominated by the faradic resistance. This reduction in impedance is result from the enhanced specific surface area of the Pt-black electrode. **d**, Optical image of flexible printed circuits (FPC) integrated with Pt-black needles ($\sim 100 \mu\text{m}$ in diameter) and $\text{Nb}/\text{NbTi}_x\text{O}_y$ memristor for HANs. **e**, Magnified image corresponding to the red box region in **(d)**. **f**, Magnified image corresponding to the red box in **(e)**.

3. Three-dimensional (3D) integration: For high-density arrays, we can explore 3D integration techniques to stack multiple HAN devices vertically, thereby increasing device density without significantly expanding the footprint (**Fig. R5d-f**). Furthermore, we can optimize the internal

structure of the HAN by integrating the liquid cell and memristor components more closely but fully isolated using vertical packaging technologies, such as Through Glass Via (TGV) or multilayer Flexible Printed Circuits (FPC). As illustrated in **Fig. R6**, a flexible HAN array has been demonstrated via the FPC integration method, where the liquid cell and memristor are isolated on opposite sides of the FPC film but electrically interconnected. These approaches will enable the development of significantly larger and denser HAN arrays, facilitating their integration into complex neuromorphic systems.

Fig. R6 | Strategies for three-dimensional integration of HANs. **a**, Schematic illustration of a proposed 3D integration scheme for HANs. An insulating plate serves as the Flexible Printed Circuit (FPC) substrate, with the solid-state memristor and electrical wiring integrated on its backside, while the electrochemical electrode is exposed on its frontside. **b**, Optical image of the FPC layer utilizing a transparent polyimide film as the insulating plate. Only 8 Au working electrodes are exposed on the front side of the FPC for the electrochemical cells as highlighted by the black dashed box. **c**, Magnified image corresponding to the red box region in **(b)**, highlighting the memristor integrated on the back side of FPC. **d**, Schematic illustrating the assembly of the FPC-integrated HAN array device. **e**, Optical image of FPC-integrated HAN array. **f**, Optical image of the flexibly integrated HAN array.

Comment 5: *On page 7, refractory-induced hyperpolarization—relevant to the prevention of over-firing in the brain—is an important inhibitory characteristic of biological neurons. A reference should be provided to help readers better understand this biological mechanism. In addition, a citation is also needed to support the statement on page 9, line 228: “It is similar to the case of biological neurons, whose firing rate would decrease for noxiously high stimuli exhibiting protective inhibition.”*

Reply 5: We have supplemented references to support these statements.

Comment 6: *In Supplementary Note 2, the explanations in line 66 (firing rate fluctuations with respect to V_{in}) and line 72 (exponential decay due to the time constant) refer to Fig. S3b and Fig. S3c, respectively. However, the figures do not appear to match the corresponding descriptions. Clarification is needed to ensure consistency between the text and the referenced figures.*

Reply 6: We have corrected this discrepancy and checked the manuscript thoroughly to ensure consistency between the text and the referenced figures.

Comment 7: Fig. S9 is not mentioned or discussed in either the main manuscript or the Supplementary Information. An explicit explanation should be provided to ensure a coherent logical flow for the reader.

Reply 7: We have incorporated an explicit explanation of Fig. S9 in the revised manuscript.

Comment 8: In Fig. 4d, h, and l, the transient firing frequency responses are presented to demonstrate the somatosensory functions of the HAN. Providing clearer labeling and more descriptive captions would enhance reader comprehension. In addition, it would be helpful to clarify the type of view (e.g., top view, cross-sectional view) presented in Fig. 5e.

Reply 8: We appreciate your suggestion. We have labeled Fig. 4d, h, and l more clearly to illustrate the operation for external environmental regulation. We have also expanded the captions of Fig. 4 and Fig. 5 to provide more detailed descriptions. Furthermore, we have clarified the type of view presented in Fig. 5e.

Comment 9: In Fig. S12, although an identical ultrasonic source is applied, the spike density appears to increase with repeated stimulation. Does the HAN maintain a constant firing rate in response to consistent input, or does this reflect a form of short-term learning behavior?

Reply 9: Thank you for your keen observation and valuable suggestion. The instability of the spike density, which appears to increase with repeated stimulation (originally depicted in Fig. S12, now Fig. R7a in this response) is due to the variation of the ultrasound source, as confirmed by a homemade piezoelectric detector. As shown in Fig. R7b, the ultrasonic source emits wavelets of obviously unstable amplitude for the first few cycles. For a stable ultrasonic source (Fig. R7c), our HAN device exhibits much more stable spikes, as shown in Fig. R7d. To avoid confusion for readers, we have replaced the spiking data with the stable one.

Fig. R7 | Ultrasonic response of the piezoelectric detector and HAN devices. a, Response of the HAN subjected to on/off cycles of ultrasonic stimuli, previously presented in Fig. S12. **b,** Unstable ultrasonic emission subjected to

on/off cycles confirmed by piezoelectric detector, exhibiting obvious variation of ultrasonic strength. **c**, stable ultrasonic emission confirmed by piezoelectric detector, exhibiting stable ultrasonic strength. **d**, Response of the HAN subjected to on/off cycles of stable ultrasonic stimuli.

Regarding the referee's insightful concern about short-term behavior, we acknowledge its importance in understanding neuronal dynamics. To investigate this, we applied a sustained train of identical input pulses to the HAN, precisely controlled by a signal generator rather than the ultrasonic stimulus. During this consistent stimulation, we observed a gradual decrease in the HAN's firing rate within 5 seconds of initial stimulation (**Fig. R8**). This phenomenon is characteristic of neuronal adaptation, where a neuron's excitability decreases under prolonged or repetitive stimulation. While adaptation represents a form of short-term neuronal dynamics, it is mechanistically distinct from conventional short-term learning behavior, which involves temporal changes in synaptic efficacy between neurons. This adaptation can be attributed to pulse-input-induced changes in the ion concentration near the electrode interface, which consequently alters the electrode potential and effective impedance, thereby diminishing the neuron's excitability. While this phenomenon merits further investigation, a comprehensive exploration of these specific dynamics falls outside the primary scope of the current work.

Fig. R8 | Short-term neuron adaptation of the HAN. **a**, Response of the HAN subjected to an input pulse train with period 0.19ms, width 0.1ms, and height 1.165 V. **b**, Evolution of the HAN's firing probability over time, calculated by statistically averaging spike occurrences within 10 ms intervals, demonstrating the gradual decrease characteristic of neuron adaptation. **c**, **d** Detailed waveform of the input voltage and current output of the HAN corresponding to (a).

Comment 10: *The caption of Fig. S15 refers to subfigures a–f; however, only a–d are shown in the figure. This discrepancy should be corrected to avoid confusion.*

Reply 10: Thank you for pointing out this discrepancy, which has been corrected.

Reply to Referee #2:

General Comments: *This study presents a hybrid artificial neuron (HAN) that integrates a solid-state memristor with an ionic electrochemical element, achieving wide frequency tunability and inherent sensory capabilities. The work addresses critical limitations of conventional solid-state artificial neurons, such as narrow frequency ranges and reliance on external sensors, by leveraging the nonlinear dynamics of ionic-electronic interfaces. The concept is innovative and holds promise for neuromorphic computing applications. However, the following points should be addressed before the manuscript can be considered for acceptance:*

General reply: Thank you for highlighting the innovation of our work. Below, we address each point in detail.

Comment 1: *The manuscript should specify the parameters of the input voltage used throughout the spiking experiments, including pulse height, interval, width, and total number of pulses.*

Reply 1: We have now explicitly detailed the parameters of the applied input voltage within the captions of Fig. 3 and Fig. S10 to ensure clarity.

Comment 2: *Please provide a detailed description of the experimental setup and methodology used to acquire the data presented in Figure 1e. Additionally, clarify the rationale for representing the time scale in milliseconds (“ms”).*

Reply 2: We have enhanced the manuscript to provide a comprehensive description of the experimental setup and methodology for acquiring the data presented in Fig. 1e. As schematically depicted in **Fig. R9**, the simultaneous voltage and current measurements were performed using an oscilloscope combined with a current amplifier. A detailed protocol has been incorporated into the Methods section and further elaborated in the Supplementary Information. Regarding the choice of a millisecond (ms) timescale, this was deliberately selected to facilitate a direct and meaningful comparison with the characteristic timescales of action potentials in biological neurons, which typically operate within this temporal range.

Fig. R9 | Experimental setup for simultaneous voltage and current measurements of the HAN (hybrid artificial neuron). Input voltage V_m is applied by a signal generator (Tektronix AFG31052). Current (I) is measured via a current amplifier (Stanford Research SR570), whose output, proportional to current, is connected to one channel of the oscilloscope. Another oscilloscope channel is directly employed to measure the memristor voltage (V_m).

Comment 3: *Figure S6 shows the voltage response across the solution during firing events. As the*

input voltage increases, both V_S^H and V_S^L increase proportionally, while the voltage difference remains nearly unchanged. The authors are asked to provide an explanation for this behavior.

Reply 3: During firing events, the memristor operates in a dynamic regime where its voltage (V_m) rapidly switches between the threshold voltage (V_{th}) and the holding voltage (V_h), as thoroughly detailed in **Supplementary Note 1**. Consequently, the highest memristor voltage (V_M^H) and lowest memristor voltage (V_M^L) during spiking activity effectively correspond to V_{th} and V_h , respectively. As confirmed by **Fig. R10a**, V_M^H and V_M^L remain remarkably constant across the tested input voltage (V_{in}) range. This intrinsic characteristic of the memristor's switching mechanism thus leads to a stable voltage difference across the memristor ($\Delta V_m = V_M^H - V_M^L = V_{th} - V_h$) that is fundamentally independent of V_{in} .

Given that the solution voltage (V_s) is determined by the fundamental circuit relationship $V_s = V_{in} - V_m$, the observed lowest solution voltage (V_s^L) becomes $V_{in} - V_M^H$, and the highest solution voltage (V_s^H) becomes $V_{in} - V_M^L$. As V_{in} increases, both V_s^L and V_s^H exhibit a proportional linear growth, as experimentally shown in **Fig. R10b**. Crucially, their difference ($\Delta V_s = V_s^H - V_s^L = V_M^L - V_M^H$) remains precisely equal to the constant ΔV_m . This elegantly explains the experimentally observed proportional scaling of V_s^H and V_s^L with V_{in} , coupled with the invariant ΔV_s , which is an inherent consequence of the memristor's well-defined threshold and holding voltages. This comprehensive explanation has been incorporated into the revised caption of Fig. S8 (previously Fig. S6).

Fig. R10 | Voltage dynamics of HAN during firing activity. **a**, Memristor voltage range (V_m) as a function of input voltage (V_{in}) during spiking. V_M^H and V_M^L represent the highest and lowest voltage divisions across the memristor during firing, respectively. Note that V_M^H and V_M^L remain nearly constant and are independent of V_{in} , corresponding to the intrinsic threshold voltage (V_{th}) and holding voltage (V_h) of the memristor. **b**, Voltage range across the redox solution (V_s) as a function of V_{in} . V_s^H and V_s^L represent the highest and lowest voltage divisions across the redox solution during firing.

Comment 4: In Figure S10, the authors present the refractory period of the HAN. However, it is unclear why a membrane potential (V_m) is observed without a corresponding current spike at $T = 0.25$ ms and 0.15 ms. Furthermore, clarification is needed regarding the initial overshoot and subsequent stabilization of V_m during the current spiking event.

Reply 4: We have inserted the voltage waveform of the input pulse (V_{in}) with a height of 1.25V into Fig. S11 (previously Fig. S10) for better context. This waveform is also presented here in **Fig. R11a and b**. When an input pulse is applied, the memristor is initially in its high-resistance state, possessing an impedance significantly greater than that of the liquid component. Consequently, it

instantaneously divides the majority of the input voltage, resulting in a distinct memristor voltage (V_m) pulse (approximately 1.17 V). If a spike is not elicited (e.g., at $T = 0.25$ ms and 0.15 ms), the memristor simply maintains this high-resistance state, and V_m retains a high voltage division dictated by the input pulse voltage. This explains the observation of a V_m pulse without a concomitant current spike.

Conversely, if a spike is elicited, the memristor undergoes a rapid transition from the high-resistance (OFF) state to the low-resistance (ON) state. This abrupt switch in resistance corresponds to the switch-on routine along the load line (**Supplementary Note 1**) as shown in **Fig. R11c**. As a result, a sharp and substantial increase in current (I) occurs, accompanied by a sudden drop in V_m from point A to B, which manifests as the initial overshoot observed during each spiking event as shown in **Fig. R11d**. Following this initial rapid drop, V_m approaches the holding voltage (V_h). However, the high differential resistance of the memristor in the vicinity of V_h induces significant decrease in current I while inducing only minor variations in V_m as shown in **Fig. R11c**. Consequently, V_m exhibits an "ultraslow" decay, appearing to stabilize after the initial overshoot, before eventually returning to its resting state upon termination of the input pulse. This detailed explanation has been incorporated into the revised caption of Fig. S11 (previously Fig. S10).

Fig. R11 | Dynamic behavior of HAN under pulse stimuli. **a**, Input voltage waveforms (V_{in}), memristor voltage responses (V_m), and current spike waveforms (I) under different stimulus intervals of 0.15 ms and **(b)** 0.6 ms. **c**, Dynamic I-V characteristic of the memristor during firing activity, originally presented in Fig. 1f in the main text. It highlights the switch-on routine labeled as 'on' from point A to B, contributing to the observed initial overshoot of I and V_m during each spiking event. **d**, Magnified waveform of V_{in} , V_m , and I during a typical spiking event. Gray dashed arrow labeled as 'on' corresponds to the transient switch-on process in (c). Bandwidth of the current preamplifier (SR 570) employed for I acquisition is 1.0 MHz.

Comment 5: Regarding the array configuration shown in Figure 5e, please specify the exact dimensions of the setup. The manuscript should also provide evidence supporting the claim that

neighboring cells do not interfere with each other, especially considering that all cells share the same electrolyte and bottom electrode.

Reply 5: We have provided the exact dimensions of the array device in Fig. S16, Fig. S17 and **Fig. R12a-c**. Regarding the potential interference between neighboring HANs due to the shared electrolyte and common ground electrode, our design strategically minimizes such coupling. The area of the shared ground electrode is significantly larger than the sum of the working electrode areas of all individual HANs, with a ratio of approximately 18. This large common electrode ensures that changes in electrical potential at any single working electrode have a negligible effect on the potential drop at the liquid-ground electrode interface, effectively isolating the individual cells. To provide direct experimental evidence for this isolation, we performed a series of activation experiments. We adjusted all six HANs in the array to a critical bias voltage just below their spiking threshold. As shown in **Fig. R12d-f**, when we selectively increased the input voltage of HAN N_{12} to elicit robust spiking, no spiking or interference was observed in any of the other five HANs. Similarly, when HAN N_{32} was activated, the other HANs, including HAN N_{12} which maintained a stable spiking rate of ~ 0.6 kHz, showed negligible perturbation. These results conclusively demonstrate that neighboring cells in our array operate independently.

Fig. R12 | Independent operation of the HAN array. **a**, Optical top view of the HAN array, illustrating six individually wired working electrodes (N_{ij} , where $i = 1, 2$ and $j = 1, 2, 3$) and a shared ground electrode (GND). The yellow region indicates the $\text{Fe}^{2+}/\text{Fe}^{3+}$ redox solution, with the same area of the shared ground electrode. **b**, Magnified image corresponding to the red dashed box in **(a)**, highlighting the working area by the gray dashed box. **c**, Optical side view of the HAN array, showing the approximate thicknesses of the bottom ITO glass (~ 1.1 mm), top ITO glass (~ 1.1 mm), and the central electrochemical cell (~ 0.1 mm). **d**, State of six resting HAN neurons at a critical voltage poised for activation. **e**, Activation of HAN N_{12} by increasing its input voltage, demonstrating no activation in other neurons. **f**, Subsequent activation of HAN N_{32} by increasing its input voltage, confirming continued independent operation of HAN N_{12} (spiking at ~ 0.6 kHz) and absence of interference in other neurons.

Comment 6: For handwriting recognition, the authors claim spike amplitude can be assigned to specific neurons as a recognition feature. However, it is unclear how the final recognition decision

is determined from the network's output. In Figure 5h, it would be helpful to explain clearly why neurons N21, N22, and N31 exhibit a broader spike frequency range compared to the other neurons.

Reply 6: We apologize for the lack of clarity regarding the amplitude coding and recognition decision. We have enhanced the main text to clarify the principle of handwriting recognition. Our recognition strategy leverages a dual-encoding approach for each neuron: its unique oscillation peak amplitude and its characteristic firing frequency range. These two features serve as orthogonal dimensions in a feature space, allowing for the precise identification of the source neuron. The peak amplitude and frequency of the output spike train are then analyzed using a DBSCAN algorithm (Supplementary Note 5). Each cluster identified by DBSCAN corresponds to a specific HAN, based on its unique amplitude and frequency signature, as depicted in Fig. 5h and Fig. S19. This clustering enables the determination of which HAN generated a particular spike sequence. The final recognition decision for a handwritten digit is then derived from the temporal sequence of these identified clusters as shown in Fig. 5g, which reconstructs the writing trajectory.

Regarding the observation that neurons N₂₁, N₂₂, and N₃₁ exhibit a broader spike frequency range compared to other neurons in Fig. 5h, this is directly attributed to their much higher firing rate. During the handwriting process, the temperature change induced by finger contact is not instantaneous. For instance, after the finger leaves a specific region, the temperature gradually decreases, causing the firing rate to consistently drop from a high level to a lower level, as shown in Fig. R13. This extended period of decreasing firing rates, rather than an abrupt cessation, manifests as a broader spectrum of observed frequencies for these sensitive neurons with much higher firing rate. We have updated Fig. S18 to better illustrate this.

Fig. R13 | a, Output waveform of the sensing array in response to the handwriting of “8”. b-d, Magnified spike waveforms extracted from (a) corresponding to electrodes N21, N22 and N31, respectively, illustrating their broader firing rate range.

Comment 7: *What is the energy consumption of the HAN system? The authors should report and compare the energy consumption of HAN with other neuron memristors.*

Reply 7: Our current HAN, considering an average firing rate of 1 kHz, consumes approximately ~1 mW. This energy consumption is predominantly determined by the ON-current of the NbTi_xO_y threshold memristor, which is typically around ~1 mA. We have now provided a comprehensive comparison of our HAN's energy consumption with other reported volatile memristors, including those based on NbO_x, HfO_x, and chalcogenide materials, in Table 1 of the Supplementary Information.

We acknowledge that the present generation of our HAN may not offer a significant advantage in energy efficiency compared to some highly optimized solid-state memristors. However, as highlighted in our reply to **Referee #1's Comment 2** and **Fig. R3**, this limitation points towards a clear optimization route. By replacing the current Nb/NbTi_xO_y memristor with alternative low-power threshold artificial neurons, such as transistor-based designs with lower ON-currents down to 2 μA as demonstrated in **Fig. R14**, we can not only maintain the wide-range frequency tunability but also significantly reduce the energy consumption.

Fig. R14 | Transistor-based HAN exhibiting widely tunable firing frequency, low power consumption, and microelectrode. **a**, Circuit schematic of transistor-based artificial neuron unit consisting of PNP and NPN transistors and two resistors. **b**, I_m - V_m characteristics of the transistor-based neuron exhibiting threshold switching behavior, similar to Nb/NbTi_xO_y memristor but with a much lower ON current of ~ 3 μ A. **c**, Schematic circuit of a HAN, composed of transistor-based neuron (purple element) in serials with a liquid cell (brown element). **d**, Spike waveforms corresponding to different input voltages V_{in} corresponding to that of matched colors in **(e)**. **e**, Log dependence of the firing frequency f on V_{in} , exhibiting a widely tunable firing frequency up to 5 orders of magnitude. **f**, Optical image of four transistor-based HANs integrated on one PCB. **g**, Magnified image of the red dashed rectangular region in **(f)**, showing the electrochemical liquid cell part of HANs with four-channel Pt electrodes and a PDMS microchannel. **h**, Magnified image of the red dashed rectangular region in **(g)**, exhibiting the Pt microelectrodes (white) and the PDMS microchannel of redox solution (Yellow).

Comment 8: *The broadly tunable frequency of HAN is fundamentally based on the controllable impedance of the Fe²⁺/Fe³⁺ redox solution. The authors should elaborate on the advantages of this redox approach and discuss its impact on the scalability and integration of HAN circuits, particularly given the complexities associated with incorporating liquid-phase redox systems in compact architectures.*

Reply 8: We agree that the controllable impedance of the Fe²⁺/Fe³⁺ redox solution is indeed central to the HAN's unique capabilities. This redox-mediated approach offers several distinct advantages

crucial for advanced neuromorphic systems. It enables an unprecedentedly wide frequency tunability spanning five orders of magnitude, inherently confers direct sensing capabilities for various physical and chemical stimuli without external transducers, and provides a biomimetic fidelity to biological ion channel dynamics.

However, we acknowledge the complexities associated with incorporating liquid-phase redox systems in compact architectures, which pose challenges for scalability and integration. We have thoroughly considered these aspects and propose several strategies to enhance both the scalability of individual HANs and the integration of their internal structure, with corresponding discussions incorporated into the main text.

1. Reducing the working current and electrode area: The working current of the HAN is primarily determined by the solid-state memristor. As demonstrated in **Fig. R14**, by replacing the NbTi_xO_y memristor with other threshold artificial neurons exhibiting lower working current such as silicon-transistor-based neurons, the working electrode size can be significantly reduced to micrometers or even sub-micrometer scales in principle. Such transistor-based neurons are highly scalable through standard CMOS processes. Besides transistors, other micro-scale and low-power threshold memristors can be potentially employed as listed in the Table 1 in the Supplementary Information. This reduction in working current directly leads to the miniaturization of the electrochemical cell.

2. Increasing the specific surface area of the electrode: We have explored methods to increase the effective specific surface area of the electrodes. For instance, electroplating platinum black (Pt-black) onto Pt electrodes significantly increased the effective surface area, leading to a ~ 2.5 -fold decrease in interfacial impedance (**Fig. R15**). This decrease in impedance directly implies that the electrode footprint can be correspondingly reduced while maintaining the same current density, thereby enabling further miniaturization.

Fig. R15 | Characterization of platinum black (Pt-black) electrodes. **a**, SEM image of Pt wire electroplated Pt-black. **b**, Magnified SEM image corresponding to the red box region in **(a)**. **c**, Bode plot of electrochemical impedance. The plot illustrates a substantial decrease in the electrochemical impedance for the Pt-black electrode compared to a bare Pt electrode of the same nominal footprint, particularly within the low-frequency region where the impedance is dominated by the faradic resistance. This reduction in impedance is result from the enhanced specific

surface area of the Pt-black electrode. **d**, Optical image of flexible printed circuits (FPC) integrated with Pt-black needles ($\sim 100\ \mu\text{m}$ in diameter) and Nb/NbTi_xO_y memristor for HANs. **e**, Magnified image corresponding to the red box region in (**d**). **f**, Magnified image corresponding to the red box in (**e**).

3. Three-dimensional (3D) integration: For high-density arrays, we can explore 3D integration techniques to stack multiple HAN devices vertically, thereby increasing device density without significantly expanding the footprint (**Fig. R15**). Furthermore, we can optimize the internal structure of the HAN by integrating the liquid cell and memristor components more closely but fully isolated using vertical packaging technologies, such as Through Glass Via (TGV) or multilayer Flexible Printed Circuits (FPC). As illustrated in **Fig. R16**, a flexible HAN array has been demonstrated via the FPC integration method, where the liquid cell and memristor are isolated on opposite sides of the FPC film but electrically interconnected through vertical vias. These approaches will enable the development of significantly larger and denser HAN arrays, facilitating their integration into complex neuromorphic systems.

Fig. R16 | Strategies for three-dimensional integration of HANs. **a**, Schematic illustration of a proposed 3D integration scheme for HANs. An insulating plate serves as the Flexible Printed Circuit (FPC) substrate, with the solid-state memristor and electrical wiring integrated on its backside, while the electrochemical electrode is exposed on its frontside. **b**, Optical image of the FPC layer utilizing a transparent polyimide film as the insulating plate. Only 8 Au working electrodes are exposed on the front side of the FPC for the electrochemical cells as highlighted by the black dashed box. **c**, Magnified image corresponding to the red box region in (**b**), highlighting the memristor integrated on the back side of FPC. **d**, Schematic illustrating the assembly of the FPC-integrated HAN array device. **e**, Optical image of FPC-integrated HAN array. **f**, Optical image of the flexibly integrated HAN array.

Comment 9: *The authors are encouraged to include TEM or SEM images to illustrate the morphology of the threshold memristor.*

Reply 9: We have now included both Scanning Electron Microscopy (SEM) and Atomic Force Microscopy (AFM) images of the memristor in Fig. S22 to illustrate the morphology of the threshold memristor. As shown in **Fig. R17**, these images provide detailed insights into the crossbar configuration, surface morphology, and layer thicknesses of the threshold memristor.

Fig. R17 | Morphological characterization of the threshold memristor. **a**, SEM image showing the crossbar configuration of the memristor device. **b**, Magnified SEM image corresponding to the red dashed box area in **(a)**. **c**, AFM image depicting the height morphology of the memristor. **d**, Magnified AFM image corresponding to the red dashed box area in **(c)**. **e**, Relative height profile corresponding to the blue dashed line in **(d)**, indicating an approximate thickness of 25 nm for the bottom Pt layer. **f**, Relative height profile corresponding to the red dashed line in **(d)**, indicating an approximate total thickness of 125 nm for the top NbTi_xO_y/Nb/Pt stack. The functional Nb/NbTi_xO_y layer has a thickness of approximately 100 nm.

Reply to Referee #3:

Comment: *I co-reviewed this manuscript with one of the reviewers who provided the listed reports. This is part of the Nature Communications initiative to facilitate training in peer review and to provide appropriate recognition for Early Career Researchers who co-review manuscripts.*

Reply: Thank you for your participation in the co-review process. We appreciate your efforts and the valuable insights.

Reply to Referee #4:

General Comments: *In this manuscript, authors developed an ion-electronic hybrid artificial neuron by compactly integrating a nonlinear electrochemical element with a solid-state memristor, which exhibits a tunable spiking frequency spanning five orders of magnitude, significantly surpassing the capability of artificial neurons based on electronic devices. This hybrid neuron design, taking advantage of both ionic and electronic features, offers a promising approach for advanced e-skin and neurointerface technologies. This is an innovative research work and the paper is well written. Therefore, I recommend that this paper can be accepted for publication after appropriate revisions.*

General Reply: We sincerely appreciate your positive assessment of our manuscript.

Comment 1: *The author should compare previous relevant reports to highlight the uniqueness of this study in the Introduction section.*

Reply 1: We agree with the referee's suggestion to highlight the uniqueness of our study more explicitly within the Introduction. While prior research has explored various forms of artificial neurons, these often exhibit limitations in terms of firing frequency tunability, requiring complex external circuitry, or lacking inherent sensory capabilities. We have revised the Introduction to include a more detailed comparative analysis.

Comment 2: *Lack of discussion on the working mechanism of the device, the author should supplement relevant discussions in the main text.*

Reply 2: We have therefore introduced a dedicated discussion to elaborate on the fundamental interplay between the electronic and ionic components that gives rise to the observed neuronal behaviors.

Comment 3: *What is the thickness of the functional layer? The author should have provided a cross-sectional SEM image.*

Reply 3: We appreciate this suggestion for providing comprehensive morphological characterization. We have included both SEM and Atomic Force Microscopy (AFM) images of the memristor in Fig. S22, also presented in **Fig. R18** to provide detailed information about the thickness and morphology of the functional layer. These images provide detailed insight into the device's layered structure and morphology. From these analyses, we can confirm that the thickness of the functional Nb/NbTi_xO_y layer is approximately 100 nm, with the top and bottom Pt electrode layers each being approximately 25 nm thick.

Fig. R18 | Morphological characterization of the threshold memristor. **a**, SEM image showing the crossbar configuration of the memristor device. **b**, Magnified SEM image corresponding to the red dashed box area in **(a)**. **c**, AFM image depicting the height morphology of the memristor. **d**, Magnified AFM image corresponding to the red dashed box area in **(c)**. **e**, Relative height profile corresponding to the blue dashed line in **(d)**, indicating an approximate thickness of 25 nm for the bottom Pt layer. **f**, Relative height profile corresponding to the red dashed line in **(d)**, indicating an approximate total thickness of 125 nm for the top NbTi_xO_y/Nb/Pt stack. The functional Nb/NbTi_xO_y layer has a thickness of approximately 100 nm.

Comment 4: *Can the author present the I-V curve of a symmetrical voltage window in the main text or supporting information.*

Reply 4: We have performed I-V measurements on a memristor under a symmetrical voltage window for 100 cycles. **As shown in Fig. R19**, it exhibited threshold-switching behavior in both the positive and negative voltage sweeping directions. This data has been supplemented in Fig. S1 to provide a comprehensive understanding of the memristor's bidirectional electrical characteristics.

Fig. R19| Bidirectional I-V characteristic of a threshold memristor under 100 cycles of symmetrical voltage sweeping.

Comment 5: *Grammar errors require careful correction by the author.*

Reply 5: Thank you for your kind reminder. We have carefully reviewed the grammar errors throughout the manuscript to ensure clarity and precision.

Reply to Referee #1:

General Comments: *The authors have mostly addressed the reviewer's concerns.*

General Reply: We sincerely thank the reviewer for the time and effort in evaluating our manuscript. We are grateful for the constructive comments provided during the review process, which have helped us improve the manuscript.

Reply to Referee #2:

General Comments: *The authors have adequately addressed all my previous comments. I have no further concerns.*

General Reply: We thank the reviewer for carefully reading our revised manuscript and for acknowledging that all previous concerns have been adequately addressed. We appreciate the constructive feedback provided throughout the review process.

Reply to Referee #3:

General Comments: *I co-reviewed this manuscript with one of the reviewers who provided the listed reports. This is part of the Nature Communications initiative to facilitate training in peer review and to provide appropriate recognition for Early Career Researchers who co-review manuscripts.*

General Reply: We appreciate your contribution to peer review and are grateful for your time and support in evaluating our manuscript.

Reply to Referee #4:

General Comments: *This manuscript can be accepted for publication.*

General Reply: We sincerely thank the reviewer for their positive assessment and recommendation for acceptance. We are encouraged by this feedback and appreciate the time taken to review our work.